# NKX2-5 Variant in Two Siblings with Thyroid Hemiagenesis

**DOI:** 10.3390/ijms23063414

**Published:** 2022-03-21

**Authors:** Ewelina Szczepanek-Parulska, Bartłomiej Budny, Martyna Borowczyk, Igor Zhukov, Kosma Szutkowski, Katarzyna Zawadzka, Raiha Tahir, Andrzej Minczykowski, Marek Niedziela, Marek Ruchała

**Affiliations:** 1Department of Endocrinology, Metabolism and Internal Diseases, Poznan University of Medical Sciences, 61-701 Poznan, Poland; bbudny@ump.edu.pl (B.B.); raiha.tahir@outlook.com (R.T.); mruchala@ump.edu.pl (M.R.); 2Department of Medical Simulation, Poznan University of Medical Sciences, 61-701 Poznan, Poland; martyna.borowczyk@ump.edu.pl; 3Polish Academy of Sciences, Institute of Biochemistry and Biophysics, 02-106 Warsaw, Poland; igor@ibb.waw.pl; 4NanoBioMedical Centre, Adam Mickiewicz University, 61-614 Poznan, Poland; kosma_sz@amu.edu.pl; 5MNM Diagnostics Sp. z o.o, 61-695 Poznan, Poland; katarzyna.zawadzka@mnm.bio; 6Department of Intensive Cardiological Care and Internal Medicine, Poznan University of Medical Sciences, 61-701 Poznan, Poland; anmin@ump.edu.pl; 7Department of Pediatric Endocrinology and Rheumatology, Institute of Pediatrics, Poznan University of Medical Sciences, 61-701 Poznan, Poland; mniedzie@ump.edu.pl

**Keywords:** thyroid hemiagenesis, thyroid dysgenesis, thyroid transcription factor, *NKX2-5* gene, whole-exome sequencing

## Abstract

Thyroid hemiagenesis (THA) is an inborn absence of one thyroid lobe of largely unknown etiopathogenesis. The aim of the study was to reveal genetic factors responsible for thyroid maldevelopment in two siblings with THA. None of the family members presented with congenital heart defect. The samples were subjected to whole-exome sequencing (WES) (Illumina, TruSeq Exome Enrichment Kit, San Diego, CA 92121, USA). An ultra-rare variant c.839C>T (p.Pro280Leu) in *NKX2-5* gene (NM_004387.4) was identified in both affected children and an unaffected father. In the mother, the variant was not present. This variant is reported in population databases with 0.0000655 MAF (GnomAD v3, dbSNP rs761596254). The affected amino acid position is moderately conserved (positive scores in PhyloP: 1.364 and phastCons: 0.398). Functional prediction algorithms showed deleterious impact (dbNSFP v4.1, FATHMM, SIFT) or benign (CADD, PolyPhen-2, Mutation Assessor). According to ACMG criteria, variant is classified as having uncertain clinical significance. For the first time, *NKX2-5* gene variants were found in two siblings with THA, providing evidence for its potential contribution to the pathogenesis of this type of thyroid dysgenesis. The presence of the variant in an unaffected parent, carrier of p.Pro280Leu variant, suggests potential contribution of yet unidentified additional factors determining the final penetrance and expression.

## 1. Introduction

Patients with thyroid dysgenesis (TD) account for 80–85% of congenital hypothyroidism (CH) cases. However, only in a small percentage of these patients, mutations in thyroid transcription factors (NKX2-1, PAX8, FOXE1, and NKX2-5) have been identified [1]. Thyroid hemiagenesis (THA) is an inborn absence of one thyroid lobe of largely unknown etiopathogenesis, affecting 0.05–0.5% of the population [2]. The anomaly rarely causes CH, thus may remain undetected until adulthood [3,4]. However, it has been demonstrated to pose an increased risk of thyroid compensatory enlargement, nodular goiter, and possibly also autoimmune thyroid diseases [2]. To date, in humans with THA, only heterozygous mutation in *PAX8*, microduplication of 22q11.2 chromosome region encompassing *TBX1* gene in one patient and several variants in *HOXB3*, *HOXD3*, *PITX2*, *GLI3* and proteasome genes were suspected to cause THA. Variations in *FOXE1* polyalanine tract in patients with hereditary form of THA have also been demonstrated [5]. In addition, an alternative 3′ acceptor site in the exon 2 of human *PAX8* gene resulting in the expression of unknown mRNA variant found in THA and some types of cancers has been demonstrated [6]. The aim of the study was to identify genetic factors responsible for thyroid maldevelopment in two siblings with THA.

## 2. Results

Patients diagnosed with THA presented neither clinical nor biochemical features of CH. The male proband was first diagnosed at the age of 10 due to suspicion of hypothyroidism, which was based on the presence of dry skin and weight gain. During physical examination bilateral ptosis was noticeable. He underwent plastic surgery of both eyelids at the age of two years, which in consequence led to lagophthalmos and periodically occurring episodes of conjunctivitis and keratitis. Each fundus of the eye had no signs of pathology.

Moreover, drooping of the left corner of the mouth and sensitization were also noticed at the time of examination (presumably due to left-sided facial nerve palsy). Since the age of seven years, he used spectacles due to slight hyperopia of the left eye (+1.0) and slight myopia of the right eye (−0.5) as well as mixed astigmatism of both eyes. There were other symptoms in his prior history such as swallowing problems (or dysphagia) and tendency to vomit, thus suggesting their neurological or muscular origin. However, the MRI of the brain reveled no important anatomical abnormalities. His weight was between the 75th and 90th centile and his height between the 3rd and 10th centile. The patient was prepubertal (axilarche 1, pubarche 1, testes volume 3 mL) and his right thyroid lobe was palpable (goiter I grade based on WHO classification). He was delivered at term and perinatal history was uncomplicated. Thyroid scintiscan (Figure 1A) and ultrasound (Figure 1B) revealed the presence of right thyroid lobe (volume 1.4 mL) and lack of the left lobe. The patient presented with hypertyreotropinaemia [TSH 5.8 µIU/mL (normal 0.27–4.2)], followed by normal TSH 3.585 µIU/mL (normal 0.470–4.640), FT_3_ 3.21 pg/mL (normal 1.45–3.48), and FT_4_ 0.76 ng/dl (normal 0.71–1.85)]. Abdominal ultrasound was normal. The dose of 12.5 µg of L-thyroxine has been started since periodic hyperthyreotropinemia and goiter were found. Audiometry performed at the age of 18 years revealed no abnormalities. Psychological examination demonstrated borderline intellectual disability, presumably due to a mild form of cerebral palsy. He finished vocational school and works as a cook’s assistant.

Proband’s sister was euthyroid [TSH 3.466 µIU/mL (normal 0.470–4.640), normal FT_3_ (2.90 pg/mL) and FT_4_ (0.95 ng/dL)], diagnosed at the age of 13 due to suspicion of goiter. She was pubertal (axilarche 3, pubarche 4, thelarche 4) and had menarche at 12 years of age. Her right thyroid was palpable (goiter I grade based on WHO classification). Thyroid scintiscan (Figure 1C) and ultrasound (Figure 1D) revealed similarly left-sided THA and normal right lobe (volume 2.52 mL). Abdominal ultrasound was normal. The dose of 25 µg of L-thyroxine was started based on goiter and ongoing puberty. She has never presented any ophthalmological nor neurological symptoms.

Anti-thyroid autoantibodies (TRAb, TPOAb, TgAb) were within normal ranges in both patients. None of the siblings nor their parents presented with congenital heart defect on echocardiography. The echocardiographic images of the male proband were deposited as Appendix A.

Family history for thyroid disorders was positive in their mother, who underwent thyroid surgery at the age of 27 years due to a toxic nodule; follicular adenoma was found on histopathology. Proband’s grandmother on mother had normal bilobed thyroid on thyroid ultrasound. The pedigree of the family is shown in Figure 2.

An ultra-rare heterozygous variant c.839C>T (p.Pro280Leu) in *NKX2-5* gene (NM_004387.4) was identified in both affected children and an unaffected father. In the mother, the variant was not present (Figure 2). This variant is reported in population databases with 0.0000655 MAF (GnomAD v3, dbSNP rs761596254). The affected amino acid position is moderately conserved (positive scores in PhyloP: 1.364 and phastCons: 0.398) as showed on Figure 3A. The conservation was also elucidated in regard of the entire human NKX2 protein family and demonstrated (Figure 3B) as comparison of regions surrounding variant p.Pro280Leu among the entire NKX2 (NKX2-1 to NKX2-5) class. Functional prediction algorithms showed deleterious impact (dbNSFP v4.1, FATHMM, SIFT) or benign (CADD, PolyPhen-2, Mutation Assessor). According to ACMG criteria, variant is classified as having uncertain clinical significance (VUS) fulfilling the following criteria: PM1 (strong, location of a hot spot/critical functional domain), PM2 (supporting, absent from controls), PP2 (supporting, missense variant in a gene that has a low rate of benign missense variation), BS2 (supportive; observed in a healthy adult individual for a AR, AD or XLR disorder, with full penetrance expected at an early age). The BS2 criterion is applied when the variant is reported in population database (7/109326 alleles in GnomAD database, European population), and is a major factor for classifying VUS instead of pathogenic/likely pathogenic. However, in case of concealed traits, it is impossible to establish a true clinical status of carriers (regarded as healthy by default). Although, referring to reported frequency of TH that is higher, the presence of an ultra-rare variant in population databases is explicable and even expected. The simulated structures obtained from homology modelling are shown in Figure 3C. The quality of models is represented by Z-score parameters directly calculated using Yasara and Yasara2 forcefield. While both models obtained fair quality Z-scores with respect to 1D packing (−1.85 for normal and −1.849 for the variant), overall quality scores were poor, mainly due to the lower scores for 3D packing. The obtained Z-scores were equal to −2.296 for normal protein and −2.342 for the variant, respectively. Overall, change in the secondary structure was most prominent in the increase of the β-sheet content from 5.9% for normal protein up to 11.1% for the variant, respectively. Accordingly, the reduction of turns and coils from 60.5% (normal) to 55.6% (variant) was also observed along with a small change of α-helix content by 0.3%. To confirm our findings, further details on WES and modelling data were provided as Appendix A.

## 3. Discussion

NKX2-5 is a homeodomain-containing transcription factor crucial for normal heart morphogenesis and function. Its expression has also been shown during early stages of thyroid development of the murine. Nkx2-5 transcript has been demonstrated in the precursors of thyroidal cells in the pharyngeal floor, while at later stages Nkx2-5 pharyngeal expression is limited to the area corresponding to the thyroid primordium [7]. Nkx2-5 knock-out embryos exhibit thyroid bud hypoplasia [8]. Mutations in this gene have been documented in patients with congenital heart disease and CH.

The presence of mutations in the *NKX2-5* was to date detected in several patients with thyroid developmental defects, namely thyroid agenesis [p.R25C], hypoplasia [c.63A>G (p.E21E), c.73C>T (p.R25C), c.293G>A (p.S98N), c.676G>A (p.D226N)], and ectopy [p.R25C, p.A119S, p.R161P]. Most of them are not associated with concomitant congenital heart defect. Functional characterization performed for some of the mentioned variants demonstrated reduced DNA binding and/or transactivation properties, with a dominant-negative effect on wild-type *NKX2-5* [8]. In addition, among children with CH, a higher incidence of congenital heart defects has been observed. Based on these findings, genes expressed during embryogenesis in precursors of thyroid and heart are promising candidate genes to be considered for their involvement in the pathogenesis of TD. The variant detected in our patients’ family has been so far reported to be causative in an individual affected with accessory atrioventricular connection and one individual referred for primary electrical disease testing [9,10]. However, neither this variant nor any other mutation in *NKX2-5* gene were so far detected in patients with THA.

Ramos et al. found one patient presenting a novel heterozygous nonsynonymous substitution, c.293G>A; p.S98N, in the *NKX2-5* gene in a patient with thyroid hypoplasia identified among Brazilian patients with CH [3]. *NKX2-5* mutational screening in 241 patients with TD allowed for identification of three heterozygous missense changes: (1) p.R25C in two patients: a 24-year-old woman with no evidence of cardiac malformation with thyroid ectopy and hypothyroidism and a 15-year-old boy with thyroid agenesis born at the 35th week of gestation. Since birth, the baby presented bilateral cortex atrophy and attention deficit hyperactivity disorder, but his intelligence quotient was normal, and no history of cardiac abnormalities was reported; (2) p.A119S in a 13-year-old girl with thyroid ectopy and hypothyroidism but no heart defect; (3) p.R161P in a six-year-old girl with thyroid ectopy and hypothyroidism. This patient exhibited patent foramen ovale at birth that resolved spontaneously. Minor mitral valve insufficiency was detected in four patients with TD. Functional characterization of the three mutations demonstrated reduced DNA binding and/or transactivation properties, with a dominant-negative effect on wild-type NKX2-5 [8]. Cerqueira et al. demonstrated a significant association of c.63A>G polymorphism with thyroid hypoplasia (*p* < 0.036). The c.541G>A variant was observed in only one patient with isolated thyroid hypoplasia [11]. Long et al. through targeted next-generation sequencing of thirteen causative genes in 106 Chinese patients with CH was able to detect one novel missense *NKX2-5* mutation (c.416G>A in exon 2, NM_001166176) in one patient with TD (type not specified) [12]. Hermanns et al. identified a new paternally inherited heterozygous mutation in the *NKX2-5* gene (p.S265R) and a new maternally inherited heterozygous mutation in the PAX8 promoter region (c.-456C>T) in a girl with TD (presumably athyreosis). Both parents and a brother, who was also heterozygous for both mutations, were phenotypically normal. Immunofluorescence microscopy showed a correct nuclear localization of both wild-type (WT) and mutant NKX2-5 proteins. EMSA demonstrated that the mutant NKX2-5 binds to the NKE_2, DIO2, TG, and TPO promoter elements equally well as the WT protein. However, the mutant NKX2-5 protein showed a 30–40% reduced transactivation of the thyroglobulin and the thyroid peroxidase promoters and a dominant-negative effect of the mutant NKX2-5. EMSA studies of the WT and mutant PAX8 promoter sequences incubated with nuclear extracts from PCCL3 cells exhibited a loss of protein binding capacity of the mutant promoter. In addition, the mutant PAX8 promoter showed a significantly reduced transcriptional activation of a luciferase reporter gene in vitro. Thus, this promoter mutation is expected to lead to reduced PAX8 expression [13]. Makretskaya et al. demonstrated a heterozygous variant of *NKX2-5* c.G676A of unknown pathogenicity and significance in a patient with thyroid hypoplasia and CH [14]. Wang et al. found one patient with a non-synonymous variant p.N291I in a patient with CH and thyroid agenesis [15]. Yu et al. diagnosed c.416G>A (p.S139N) variant in one Chinese patient with CH [16]. Khatami et al. studied 65 patients with CH, thyroid hypoplasia, and no cardiovascular defects. They identified two known variations c.73C>T (p.R25C) and c.63A>G (p.E21E) of the gene. Both of them are located in the conserved region of the gene and were previously reported in cases with TD and congenital heart defects. There was a significant association between 63A>G variation with primary hypothyroidism (*p* = 0.003). These alterations are probably related to thyroid hypoplasia because the allele frequency of the c.63A>G variant was significantly different in patients and controls, and also p.R25C variation was not observed in control cases [17].

Despite reports of four subjects presenting heterozygous loss-of-function *NKX2-5* mutations associated with thyroid ectopy or agenesis, the role this gene in the development of TD remains to be elucidated [18]. Inheritance in these families was autosomal dominant, as it was in our patients; however, mutation penetrance was highly variable (carrier parents frequently had properly developed thyroid gland). In addition, the diagnosed mutations were also detected in healthy subjects (p.R25C, MAF > 1%) or were found to be associated with isolated congenital heart defects [8,19]. Thus, it has been postulated that *NKX2-5* variants may be causative for thyroid developmental abnormalities; however, there are other factors or genes that are necessary for the appearance of phenotype, presumably by modulation of expression and penetrance [18]. The modelled multidomain 3D structure of the NKX2-5 protein comprised central DNA-binding domain and facilitated recognition of the specific DNA sequence during G1 phase of the cell cycle. Substitution of serine 280 to leucine is located in the domain, which is responsible for the interaction with other proteins involved in the thyroid development pathway. Consequently, the change points to the important Ser-Pro peptide bond. This suggests the ability of serine phosphorylation at position 280 in the wild-type NKX2-5, selecting the peptidyl-prolyl cis/trans isomerases (PPIases) as a possible binding partner for the signal transduction. It is well documented that Pin1 protein controls processes at the cell cycle G1/S check-point via cis/trans isomerization Ser/Thr-Pro peptide bond [20]. This process is strictly regulated by the CDK (Cyclin-Dependent Kinase) partners [21]. In particular, we suppose that Pin1 through CDK2/4 complex control phosphorylation of the transcription factors belonged to the FOX family [22].

Ophthalmological and neurological symptoms could spread a new light on the role of detected variant in the proband. Even though the WES examination is nowadays regarded as an efficient and reliable approach for detecting alterations in coding sequences, the pathogenic variants in regulatory regions of other genes cannot be ruled out. Particularly, concerning the unusual phenotypical presentation in the male patient, with associated neurological symptoms. Likewise, no other genetic evidence of causativeness was established (i.e., oligogenicity), but more complex background contributing to the proband’s clinical status is possible. An important limitation of our study is also lack of functional studies that would confirm significance of detected NKX2-5: p.P280L variant. In this regard, the unequivocal conclusion on causativeness of the variant cannot be made and further studies are required to fully support pathogenicity.

## 4. Material and Methods

We evaluated clinically and genetically two siblings with THA and their parents. The measurement of TSH and thyroid hormones (TSH, FT_3_, FT_4_) were performed with the use of a Hitachi Cobas e601 chemiluminescent analyzer (Roche Diagnostics, CH-4070, Basel, Switzerland). The concentrations of antithyroid autoantibodies (antithyroid peroxidase antibody—TPOAb, antithyroglobulin antibody—TgAb, thyrotropin receptor antibody—TRAb) were assessed by radioimmunological method with the use of commercially available BRAMHS anti-TPO, anti-Tg, and TRAK RIA kits, by scintillation gamma counter (LKB Wallac CliniGamma 1272). Thyroid ultrasound examinations were performed with the use of 17 MHz linear probe and the ALOKA SSD 3500 SV equipment. Tc-99m thyroid scintiscan was done with the use of The GE Infinia Hawkeye 4 gamma camera.

### Genetic Examination

Blood samples were collected from all family members presented on the pedigree. Genomic DNA was extracted from peripheral blood leukocytes according to standard procedures (GTC method). We performed targeted capillary sequencing of known genes contributing to thyroid embryogenesis, namely *TTF1, TTF2, PAX8, HHEX, SHH, TBX1*. Mutation scanning was carried out by PCR reaction and following sequencing of the PCR products. The entire coding sequence of *TTF1, TTF2, PAX8, HHEX, SHH,* and *TBX1* was amplified, as well as neighboring exons intronic sequences (min. 100 nucleotides of each intron—exon boundary). Primer sequences were designed using Primer3 algorithm (http://bioinfo.ut.ee/primer3-0.4.0/, accessed on February 2022). Genomic sequences were retrieved from GeneBank. Amplifications were conducted with the use of FailSafe™ PCR PreMix J System (Epicentre, Illumina San Diego, CA 92121, USA) and processed through 40 cycles (30 s at 94 °C, 30 s at 62 °C, and 45 s at 72 °C). Sanger sequencing was conducted on both strands with the use of BigDye chemistry version 3.1, on an ABI 3130xl DNA Analyzer (ThermoFisher Sci. Foster City, CA, USA). Samples were processed through 30 cycles of amplification consisting of 30 s at 94 °C, 30 s at 60 °C, and 45 s at 72 °C. The final step was lengthened to 7 min. Analysis of sequence tracks was achieved with the use of CodonCode Aligner software version 4.0.4 (www.codoncode.com).

No mutations were found in this approach and, therefore, samples were subjected to whole-exome sequencing (WES) (Illumina, TruSeq Exome Enrichment Kit, San Diego, CA 92121, USA). A total of 1μg of genomic DNA from subjects was used for the construction of a library with the TruSeq DNA Sample Preparation Kit (Illumina). Whole exome enrichment was performed with the use of DNA library and TruSeq Exome Enrichment Kit (Illumina). The achieved assay performance was as follows: the minimal mean depth of target regions was 243X (post-alignment) and the average read length 150 bp. The percentage of bases in target regions with a depth of coverage >20X—97.9%, >30X—97.1%, >50x—93.8%. 99% of the reads was aligned and mapped to human genome reference sequence hg19 (BWA v0.7.12, Picard v1.130, GATK v3.4.0, SnpEff v4.1g), of which 90.5% were non-redundant.

For pathogenicity evaluation Sift, PolyPhen2, dbNSFP, FATHMM, MutationTaster v2, PhenIX, HPO database, and population data (GnomAD v2.1.1) were used. ACMG classification was accomplished using Varsome and InterVar.

The proposed model structure of NKX2-5 protein was obtained through homology modelling using Yasara 21.6.17 software [23,24]. Consecutive homology models were built using PsiBLAST search algorithm [25], and UniRef90H structure database (UniProt) [26], directly from FASTA file format. The model quality was checked using Yasara Z-score parameters [27]. The models were aligned using MUSTANG algorithm [28].

## 5. Conclusions

For the first time, *NKX2-5* gene variants were found in two siblings with THA, providing evidence for potential contribution of genetic background to the pathogenesis of this type of TD. The presence of the change in an unaffected parent suggests possible influence of unidentified additional factors impacting the penetrance and expressivity.

## Figures and Tables

**Figure 1 ijms-23-03414-f001:**
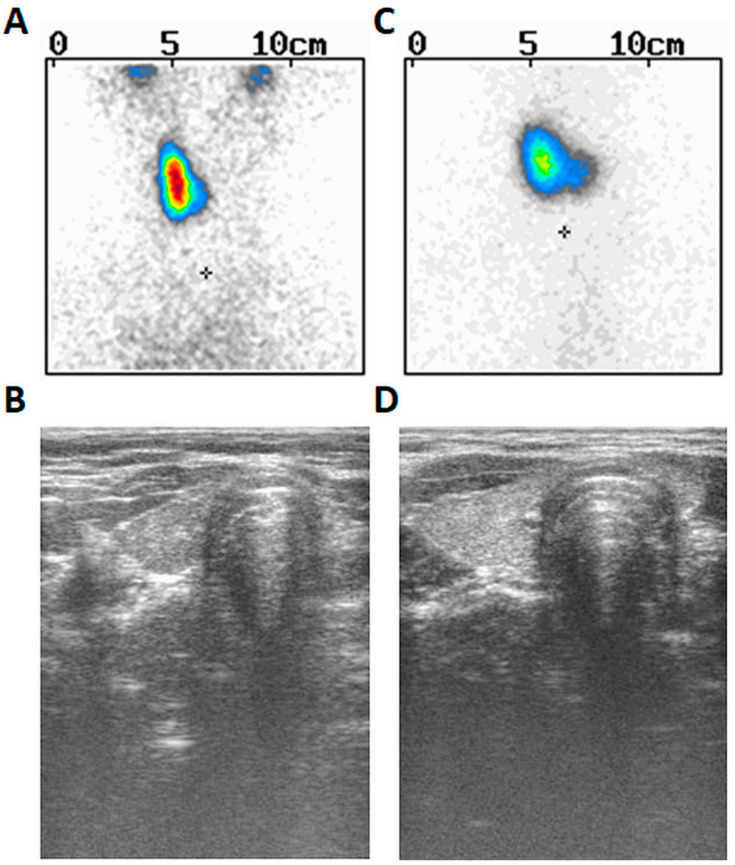
The thyroid scintiscan and ultrasound picture of both siblings with left-sided thyroid hemiagenesis. (**A**)—thyroid Tc-99m scintiscan of the male proband. The “+” in the image corresponds to the jugular notch of the sternum. (**B**)—thyroid ultrasound of the male proband. (**C**)—thyroid Tc-99m scintiscan of the male proband’s sister. The “+” in the image corresponds to the jugular notch of the sternum. (**D**)—thyroid ultrasound of the male proband’s sister.

**Figure 2 ijms-23-03414-f002:**
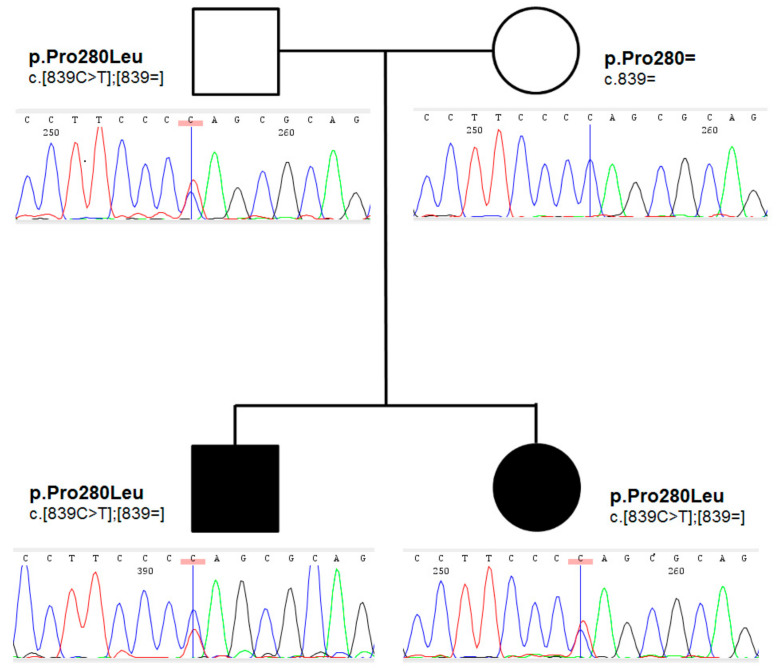
Abbreviated pedigree of the family under study. Symbols are presenting phenotypical manifestation (affected individuals are presented using shaded symbols). Below the symbols, sequencing chromatograms presenting genetic status (zygosity), and HGVS description of corresponding genotypes on protein and coding level, were presented.

**Figure 3 ijms-23-03414-f003:**
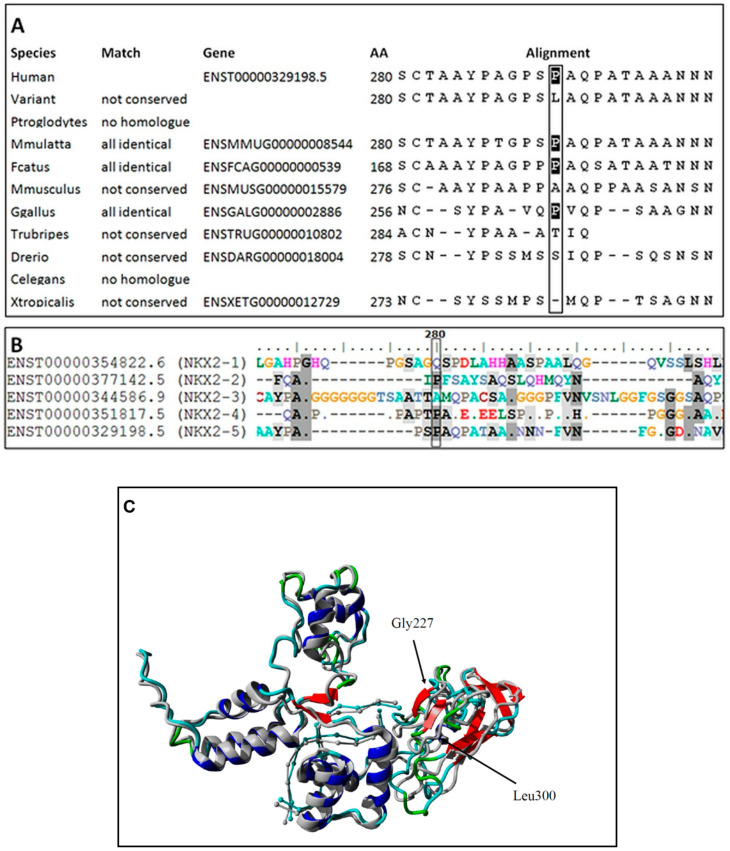
Evolutionary conservativeness of sequence surrounding variant p.Pro280Leu (**A**), comparison of sequence surrounding variant p.P280L among entire class of human *NKX2* genes (NKX2-1 to NKX2-5) (**B**) and in silico prediction of conformational change caused by a p.Pro280Leu substitution (**C**) in two predicted structures for NKX2-5 protein. The normal protein is depicted in gray, while the impact of mutation is shown in color.

## Data Availability

The data presented in this study are available on request from the corresponding author.

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
