# Peer review of "NKX2-5 Variant in Two Siblings with Thyroid Hemiagenesis"

_ijms, 2022, doi:10.3390/ijms23063414_

Round 1

Reviewer 1 Report

NKX2-5 variant in two siblings with thyroid hemiagenesis

Szczepanek-Parulska et al. comunicated the identification of the variant c.839C>T (p.Pro280Leu) in the NKX2-5 gene in two siblings with thyroid hemiagenesis in the absence of congenital heart defects. The mentioned variant was identified in heterozygosis using whole-exome sequencing. However, Sanger sequencing data supporting the identification of the variant is not available.

The evidence presented by the authors is incomplete in order to support the pathogenicity of the variant as no functional characterization of the variant is available. The variant was identified in the affected siblings but also in the unaffected father, suggesting a potential contribution of yet unidentified additional factors determining the penetrance of the variant.

The pathogenicity of the variant was assessed using in silico analysis, reaching conflicting results. The authors should have considered that prediction algorithms usually fail to accurately predict pathogenicity in sequences with low conservation. Therefore, as indicated in the previous paragraph, functional data analysis is encouraged. Moreover, the authors simulated the structure of NKX2-5 using homology modelling based on an unmentioned template. Otherwise NKX2-5 homology model construction is available, the data is unreliable.

The overall presentation of the manuscript should be improved. The clinical case should be presented in the results section. Material and methods should be completed as most of the methods has not been described.

Author Response

Szczepanek-Parulska et al. comunicated the identification of the variant c.839C>T (p.Pro280Leu) in the NKX2-5 gene in two siblings with thyroid hemiagenesis in the absence of congenital heart defects. The mentioned variant was identified in heterozygosis using whole-exome sequencing. However, Sanger sequencing data supporting the identification of the variant is not available.

Re: The Sanger sequencing plots were added to figure presenting pedigree. This figure was moved to results section (suggestion of the second reviewer) and renumbered accordingly. Also raw sequencing plots (.ab1 files) are available if necessary. Description of alteration (HGVS desc.) was also added to this figure so the zygosity status of each presented individual is now clear.

The evidence presented by the authors is incomplete in order to support the pathogenicity of the variant as no functional characterization of the variant is available. The variant was identified in the affected siblings but also in the unaffected father, suggesting a potential contribution of yet unidentified additional factors determining the penetrance of the variant. The pathogenicity of the variant was assessed using in silico analysis, reaching conflicting results. The authors should have considered that prediction algorithms usually fail to accurately predict pathogenicity in sequences with low conservation. Therefore, as indicated in the previous paragraph, functional data analysis is encouraged.

Re: Indeed, a functional examination is always considered as an experiment that more reliably reflecting cellular processes in patient. Although in this case we cannot afford such experiments. Therefore we did everything we could  including WES in all individuals looking for possible causes related to coding sequence in mapped genes. We also are aware that alterations outside exome are possible and may impact penetrance – as stand alone or via NKX2-5 interaction as well as non-genetic, environmental factors. We also believe that the best functional models were already created – patients sharing similar genetic background and phenotype, but in order to track such variability, such rare cases should be reported. We included WES data (list of identified variants) as supplementary files to allow accessibility and further elucidation together with incoming new data from THA patients.   

Moreover, the authors simulated the structure of NKX2-5 using homology modelling based on an unmentioned template. Otherwise NKX2-5 homology model construction is available, the data is unreliable.

Re: The modelling data are included as a supplementary files. Template information as well as pdb files were given.

The overall presentation of the manuscript should be improved. The clinical case should be presented in the results section. Material and methods should be completed as most of the methods has not been described.

Re: The Materials and Methods section was completed and described in details. We added section for conventional sequencing and WES. The clinical case report details were moved to the results section.

Reviewer 2 Report

Dear Authors,

the study was well planned and performed and all sessions were dealt with completely and exhaustively. The language was clear. Obtained results are valuable and expand the knowledge of the scientific community regarding topics of interest such as the genetic cause of thyroid hemiagenesis, rarely causing congenital hypothyroidism, but increasing the risk of thyroid compensatory enlargement, nodular goiter and autoimmune thyroid diseases.

However, before the publication some minor criticisms should be addressed and in particular:

1) The format should follow the suggestions of the journal’s Instructios for Authors, with the Materials and Methods section reported before the Results section.

2) Description of panel B of Figure 1 is lacking in Results section.

3) A more detailed description of the laboratory used methods (capillary sequencing and whole-exome sequencing) would be appreciated.

4) It’s never specified all over the text the sample type (I can only image that it’s genomic DNA from lymphocytes, but this must be written in the text) and if P280L mutation is homozygous or heterozygous in the siblings and the father. Please specify the nature of the mutation.

5) A small mention of the methods used for the various hormonal dosages would be appreciated.

6) The presence of the patient's photo (Figure 3) is completely irrelevant and, although consent has been given to publication, it should be removed. It is sufficient the description of patient’s face reported in Materials and Methods.

7) The family pedigree reported in Figure 2E must be moved in the Results section.

8) The number of  self-citations must be reduced.

9) It’s highly unlikely that mutation P280L is the only one identified by exome analysis. There were further variants identified by the WES analysis? Was only the coding sequence of each gene studied or were the 5 'and 3' UTR regions also included? Please argue this aspect of the study.

10) Possible typing errors must be corrected all over the text.

Author Response

Dear Authors,

the study was well planned and performed and all sessions were dealt with completely and exhaustively. The language was clear. Obtained results are valuable and expand the knowledge of the scientific community regarding topics of interest such as the genetic cause of thyroid hemiagenesis, rarely causing congenital hypothyroidism, but increasing the risk of thyroid compensatory enlargement, nodular goiter and autoimmune thyroid diseases.

However, before the publication some minor criticisms should be addressed and in particular:

1) The format should follow the suggestions of the journal’s Instructios for Authors, with the Materials and Methods section reported before the Results section.

Re: The order of appearance of sections in the manuscript has been changed accordingly.

2) Description of panel B of Figure 1 is lacking in Results section.

Re: Figure 1 is reordered for Figure 3. The Figure 3B presenting comparison of region surrounding variant p.Pro280Leu among entire class of human NKX2 genes (NKX2-1 to NKX2-5) was added to results description.

3) A more detailed description of the laboratory used methods (capillary sequencing and whole-exome sequencing) would be appreciated.

Re: The detailed description of both methods were added to Methods section.

4) It’s never specified all over the text the sample type (I can only image that it’s genomic DNA from lymphocytes, but this must be written in the text) and if P280L mutation is homozygous or heterozygous in the siblings and the father. Please specify the nature of the mutation.

Re: All informations were added – both phenotypical and genetic status of examined family members. The detailed information describing this issue was also included on the Figure presenting pedigree, together with sequencing chromatograms and HGVS description. Specimen type was also clarified and included in Methods

5) A small mention of the methods used for the various hormonal dosages would be appreciated.

Re: The details of the hormonal assessment and thyroid imaging were provided in the material and methods section.

6) The presence of the patient's photo (Figure 3) is completely irrelevant and, although consent has been given to publication, it should be removed. It is sufficient the description of patient’s face reported in Materials and Methods.

Re: The patient’s face photo has been removed according to the suggestion.

7) The family pedigree reported in Figure 2E must be moved in the Results section.

Re: This figure was moved to results section and renumbered accordingly. The Sanger sequencing plots were added to figure presenting pedigree. Description of alteration (HGVS desc.) was also added to this figure so the zygosity status of each presented individual is now clear.

8) The number of  self-citations must be reduced.

Re: The number of self-citations has been reduced.

9) It’s highly unlikely that mutation P280L is the only one identified by exome analysis. There were further variants identified by the WES analysis? Was only the coding sequence of each gene studied or were the 5 'and 3' UTR regions also included? Please argue this aspect of the study.

Re: Yes. The complete WES data of family covering high and moderate impact variants is included in supplementary file. Yes, entire transcripts were examined.

10) Possible typing errors must be corrected all over the text.

Re: The manuscript was evaluated carefully and typing errors have been corrected.

Round 2

Reviewer 1 Report

I reinforce my previous comment "The evidence presented by the authors is incomplete in order to support the pathogenicity of the variant as no functional characterization of the variant is available".

However, I understand that functional characterization is not available. Therefore, I strongly recomment to include a comment in the discussion indicating that functional characterization of the variant remains as a open question to fully support pathogenicity.

Author Response

Yes, we fully agree with reviewer’s suggestion. Missing functional data constitutes an important limitation of our study, which does not allow us to draw unequivocal conclusion in terms of causality. Hence, taking into account the Reviewer’s comment, we have included a strong statement at the end of the discussion section that points to this limitation, indicating that the functional characterization needs to be done before final conclusions on the pathogenicity of the variant can be made.

Following statement was added:

“An important limitation of our study is also lack of functional studies that would confirm significance of detected NKX2-5: p.P280L variant. In this regard, the unequivocal conclusion on causativeness of the variant cannot be made and further studies are required to fully support pathogenicity”.

This manuscript is a resubmission of an earlier submission. The following is a list of the peer review reports and author responses from that submission.

Round 1

Reviewer 1 Report

The protein-coding gene NKX2-5 (NK2 Homeobox 5) is located in the NK2 Family. NKX2-5 is linked to diseases such as atrial septal defects. Related pathways include Mesenchymal Stem Cells and Lineage-specific Markers, as well as NFAT and Cardiac Hypertrophy.

NKX2-5 gene has been shown for the first time to play a role in thyroid gland development by Szczepanek-Parulska et al. Authors found NKX2-5 variants in two siblings with THA suggesting that this gene may be involved in the pathogenesis of this type of thyroid dysgenesis. This variant is reported in population databases with 0.0000655 MAF (GnomAD v3, dbSNP rs761596254). The affected amino acid position is moderately conserved (positive scores in PhyloP: 1.364 and phastCons: 0.398). Having the variant in an unaffected parent implies that there may be other unidentified factors contributing to the penetration and expressivity of the variant. 

In my opinion, this is a remarkable report to be published in the IJMS as the authors present evidence on the NKX2-5X2-5 gene contribution to the pathogenesis of this type of thyroid dysgenesis. I enjoyed reading the manuscript, and after considering the following minor concerns, I strongly encourage the editor to accept it.

  1. It would be really helpful if authors could show the patient's heart scanning images in supplementary.
  2. Were patients subjected to any neurological testing, as discussed elsewhere (e.g. https://www.intechopen.com/chapters/37925).
  3. Is it possible to present a patient's diagnostic picture only after the patient has given consent?
  4. I encourage the authors to correct typos and spelling errors throughout the manuscript.

E.g., It should be ‘our patients' family’, but not ‘our family’. 

Reviewer 2 Report

In the presented manuscript the authors try to search for genetic factors responsible for thyroid maldevelopment in two siblings with thyroid hemiagenesis THA using exome sequencing.

An ultra-rare variant c.839C>T (p.Pro280Leu) in NKX2-5 gene was identified in both affected children and an unaffected father and classified as having uncertain clinical significance.

In the results chapter, the authors present the results of the pathogenicity assessment of the identified variant in silico and nothing else indicating the pathogenicity of this variant. The data presented in the article is insufficient to establish the connection of the identified mutation with the disease under study. The authors themselves identified their find as VUS. Thus, the data obtained are not of interest and novelty in the present form.